# Yeast facilitates the multiplication of *Drosophila* bacterial symbionts but has no effect on the form or parameters of Taylor's law

**Robin Guilhot**[1]*, **Simon Fellous**[1], **Joel E. Cohen**[2,3,4]

**1** CBGP, INRAE, CIRAD, IRD, Montpellier SupAgro, Univ Montpellier, Montpellier, France, **2** Laboratory of Populations, Rockefeller University, New York, New York, United States of America, **3** Earth Institute and Department of Statistics, Columbia University, New York, New York, United States of America, **4** Department of Statistics, University of Chicago, Chicago, Illinois, United States of America

* guilhoro@gmail.com

**Data Availability Statement:** The dataset is available in the open data repository Zenodo (DOI: 10.5281/zenodo.3628674).

## Abstract

Interactions between microbial symbionts influence their demography and that of their hosts. Taylor's power law (TL)–a well-established relationship between population size mean and variance across space and time–may help to unveil the factors and processes that determine symbiont multiplications. Recent studies suggest pervasive interactions between symbionts in *Drosophila melanogaster*. We used this system to investigate theoretical predictions regarding the effects of interspecific interactions on TL parameters. We assayed twenty natural strains of bacteria in the presence and absence of a strain of yeast using an ecologically realistic set-up with *D. melanogaster* larvae reared in natural fruit. Yeast presence led to a small increase in bacterial cell numbers; bacterial strain identity largely affected yeast multiplication. The spatial version of TL held among bacterial and yeast populations with slopes of 2. However, contrary to theoretical prediction, the facilitation of bacterial symbionts by yeast had no detectable effect on TL's parameters. These results shed new light on the nature of *D. melanogaster*'s symbiosis with yeast and bacteria. They further reveal the complexity of investigating TL with microorganisms.

## Introduction

Animals and plants are often associated with several symbiotic microorganisms, the interactions of which affect the ecology and evolution of hosts and symbionts alike [1, 2]. Encompassing the diversity of symbiotic communities is challenging. However, experimental systems of modest complexity enable the investigation of microbial interactions, their mechanisms and consequences for host phenotypes [3, 4] and the dynamics of microbial symbionts [5]. Conceptual tools developed by ecologists for the study of population and community ecology may be used to unveil general processes at play in symbiotic communities [1, 6].

Among these tools, Taylor's law (TL) may be used to investigate the spatial and temporal distributions of symbiotic populations of microorganisms. TL asserts that the variance of

**Funding:** This project was supported by French National Research Agency through the 'SWING' project (ANR-16-CE02-0015).

**Competing interests:** The authors have declared that no competing interests exist.

population density (or size) is a power function of the mean population density (or size) across a set of replicate populations [7]. This power-law relationship is equivalent to a linear relationship on log-log coordinates: *log(variance of population density) = a + b.log(mean of population density)*. A spatial TL holds when the mean and the variance are calculated over populations that differ in spatial location and the variance is a power function of the mean. A temporal TL holds when the mean and the variance are calculated over different points in time for each population and the variance is a power function of the mean. Many other variants of TL exist. TL holds in populations of various organisms [8] including bacteria [9, 10] and can be of interest to address questions and resolve issues in conservation biology [11], epidemiology [12, 13], human demography [14, 15], fisheries [16], forestry [17], and crop protection [18]. Whether the relationship between population means and variances follows TL, and TL's parameters when TL holds, can shed limited light on the demographic processes in the populations studied [8, 19]. For example, a TL's slope *b* that falls in the range from 1 to 2 and differs from 2 may be due to interspecific interactions within ecological communities such as competition, predation or parasitism [8, 20].

In this study, we investigated whether the spatial TL describes well the relation of the variance to the mean of population density in experimental populations of bacterial symbionts of *Drosophila;* and tested whether the addition of another *Drosophila* symbiont–a yeast–in the system affects TL. *Drosophila melanogaster* larvae rely on both bacteria and yeast for larval development [21, 22]. However, although interactions between microorganisms associated with *Drosophila* flies have been reported [4], the pervasiveness and nature of these relationships in the wild remains unclear [22, 23]. So far, no study has investigated the nature of yeast-bacteria interactions when associated to *Drosophila* larvae in ecologically realistic conditions using numerous strains freshly isolated from the wild. Our study hence aimed at describing and understanding the numerical effects of symbionts on each other in a context relevant to natural *Drosophila* biology. We assayed twenty bacterial strains in the absence and in the presence of a wild *Hanseniaspora uvarum* yeast strain. The growth of these microorganisms was studied in association with the larvae of a wild *D. melanogaster* population reared in natural fruit.

## Material and methods

### Biological material

We used twenty bacterial strains that were isolated from wild adult *Drosophila* and fruit homogenates collected in Montpellier (SF's garden and Montpellier SupAgro campus) and in Montferrier-sur-Lez (private property), southern France, except for three *Acetobacter* and *Lactobacillus* strains (Table 1). The yeast strain *Hanseniaspora uvarum* Dm6y (MN684824) was isolated from a wild *D. melanogaster* fly collected in Montpellier (SF's garden). Most of these microbial taxa had previously been identified as associated with *Drosophila* [24–26]. All field-isolated microorganisms were cultured a single time in the laboratory and stored in sterile Phosphate-Buffered Saline (PBS) solution (20% glycerol) at -80˚C until they were used in experiments. This ensured minimal adaptation to the laboratory of the tested microorganisms. The *Drosophila melanogaster* population was established from a few dozen wild individuals collected in Montferrier-sur-Lez (private property) about a year before the experiment. Conventionally reared flies had been maintained on a carrot-based laboratory medium (11.25 g.L$^{-1}$ agar, 37.5 g.L$^{-1}$ sugar, 15 g.L$^{-1}$ corn meal, 37.5 g.L$^{-1}$ dried carrot powder (Colin Ingredients SAS), 22.5 g.L$^{-1}$ inactive dry yeast, 5 ml.L$^{-1}$ propionic acid, 3.3 g.L$^{-1}$ nipagin, 25 ethanol ml.L$^{-1}$). All biological samples were collected with the permission of the private owners and the Montpellier SupAgro administration.

**Table 1. Bacterial strains used in this study.**

| Strain | Species | Origin | GenBank accession number | Type of agar plate | Temperature of incubation |
|---|---|---|---|---|---|
| R3b | *Gluconobacter* sp. | Grape berry | not referenced | MRS | 24°C |
| R6b | *Staphylococcus* sp. | Grape berry | not referenced | TCS | 24°C |
| R8b | *Gluconobacter thailandicus* | Grape berry | not referenced | TCS | 24°C |
| Dm2b | *Yersinia* sp. | *D. melanogaster* | not referenced | TCS | 24°C |
| Dm5b | *Gluconobacter* sp. | *D. melanogaster* | not referenced | TCS | 24°C |
| Dm6b | *Escherichia coli* | *D. melanogaster* | not referenced | TCS | 24°C |
| Dm8b | *Enterobacter* sp. | *D. melanogaster* | not referenced | TCS | 24°C |
| Dm10b | *Erwinia* sp. | *D. melanogaster* | not referenced | MRS | 24°C |
| Dm11b | Enterobacteriaceae | *D. melanogaster* | not referenced | MRS | 24°C |
| Ds3b | *Serratia fonticola* | *D. suzukii* | not referenced | TCS | 24°C |
| Ds4b | *Gluconobacter kondonii* | *D. suzukii* | not referenced | TCS | 24°C |
| Ds6b | *Lelliottia jeotgali* | *D. suzukii* | not referenced | TCS | 24°C |
| Ds9b | *Erwinia injecta* | *D. suzukii* | not referenced | TCS | 24°C |
| Ds10b | *Lelliottia jeotgali* | *D. suzukii* | not referenced | TCS | 24°C |
| Ds25b | *Lelliottia jeotgali* | *D. suzukii* | not referenced | MRS | 24°C |
| Ds27b | *Serratia liquefaciens* | *D. suzukii* | not referenced | MRS | 24°C |
| Ds28b | *Lelliottia* sp. | *D. suzukii* | not referenced | MRS | 24°C |
| Lp[WJL] | *Lactobacillus plantarum* | *D. melanogaster* | EU096230 [27] | MRS | 35°C |
| Lb[WJL] | *Lactobacillus brevis* | *D. melanogaster* | EU096227 [27] | MRS | 35°C |
| Ap[WJL] | *Acetobacter pomorum* | *D. melanogaster* | EU096229 [27] | MAN | 35°C |

## Experimental design

Each experimental unit consisted of a halved grape berry that we first surface-sterilized following the procedure of Behar et al. [28] and embedded in jellified purified water (6 g.L$^{-1}$ agar) in a small petri dish. This protocol ensures the removal of all microorganisms present at the surface of fruit, but does not eliminate possible microbial endophytes [29]. Our results hence reflect the biology of symbiotic bacteria and yeast in conditions comparable to those of the field, but not in sterile medium. We manually deposited fifteen fly eggs on each half grape berry. The eggs had been laid by groups of *D. melanogaster* females offered jellified grape juice plates (300 ml.L$^{-1}$ grape juice, 6 g.L$^{-1}$ agar) supplemented with cycloheximide (1 mg.L$^{-1}$) to inhibit yeast growth, and chloramphenicol (10 mg.L$^{-1}$) to inhibit bacterial growth. Repeated assays showed eggs produced in this manner are free of culturable microorganisms. After egg deposition, 10$^5$ cells of each bacterium were inoculated to fruit flesh either alone (seven replicates per bacterium, except for strain R6b that had six replicates) or together with 10$^5$ cells of the yeast strain (seven replicates per bacterium). The experiment was spread into seven blocks: one replicate of each bacterium × yeast combination was set up each day over seven days. Experimental units were incubated at 25°C.

To measure microbial growth, we sampled fruit flesh after three days of incubation in a non-destructive fashion (analyses of *D. melanogaster* development will be described in a separate manuscript). Fruit flesh was sub-sampled by randomly inserting ten sterile pipette tips in the surface of each fruit, collecting approximately one twentieth of the flesh in total. Flesh samples were pooled per replicate, homogenized in 100 µl of sterile PBS solution and serially diluted. Cell counts were carried out by plating samples (serially diluted) on appropriate selective agar media (Table 1). For bacterial detection, we used Trypto-Casein-Soy (TCS) agar, Mannitol (MAN) agar or De Man, Rogosa and Sharpe (MRS) agar, all supplemented with cycloheximide (1 mg.L$^{-1}$). For yeast detection, we used Yeast Extract-Peptone-Dextrose (YPD)

agar supplemented with chloramphenicol (10 mg.L$^{-1}$). Microbial colonies were counted after four days of incubation at species-specific temperatures and growth media (Table 1). We are confident the bacteria counted at the end of the experiment were those inoculated at the beginning for two reasons. First, we observed no bacterial growth in bacteria-free controls, which rules out the presence of culturable endophytes or contaminants in the fruits used for the experiment. Second, lack of cross-contaminations between bacterial treatments was attested by the systematic match between the morphology and metabolic abilities of the bacterial strains inoculated and those of the counted cells. These comparisons were possible because the majority of the strains used could be discriminated on the basis of the temperature and medium that enabled colony growth, as well as colony color, shape and transparency.

## Statistical analyses

Analyses were split in two steps. First, we investigated the effects of each type of symbiont, and their interactions, on mean cell numbers. We hence tested whether yeast affected (i.e. increased or decreased) bacterial densities using microbial counts from each replicate as datapoints. We used a linear mixed model with the restricted maximum likelihood method. '*Modality*' (i.e. presence / absence of yeast), '*bacterial strain*' and their interaction were defined as fixed factors; '*experimental block*' was defined as a random factor. A similar analysis was also carried out to test whether bacteria had variable effects on yeast growth. Finally, we tested whether numbers of yeast and bacterial cells correlated among bacterial treatments. To this end we used a symmetrical Major Axis regression with mean cell numbers per treatment as datapoints.

In a second stage, we investigated the variance of microbial cell numbers across twenty bacterial strains. We hence tested whether the relationship between the means and variances of bacterial densities followed a mixed-species power law (i.e. $\log_{10}$(variance) = $a$ + $b$.$\log_{10}$(mean)) [8] and whether its parameters $a$ and $b$ depended on yeast presence. We hence used a linear model with '*modality*', '*$\log_{10}$(mean)*' and their interaction as fixed factors. We further included in early model formulations the quadratic term '$[\log_{10}(mean)]^2$' and its interaction with '*modality*' to investigate deviation from a power law. A similar analysis was carried out to test whether the relationship between the means and variances of densities of the yeast *Hanseniaspora uvarum* in presence of the twenty bacterial strains followed a single-species power law (i.e. $\log_{10}$(variance) = $a$ + $b$.$\log_{10}$(mean)). We used a linear model with '*$\log_{10}$(mean)*' as a fixed factor. We further included in early model formulations the quadratic term '$[\log_{10}(mean)]^2$' to investigate deviation from a power law. We found no significant evidence of nonlinearity in the two relationships.

Analyzes were performed with JMP (SAS, 14.1) and R (3.6.2) (package lmodel2 [30]).

## Results

### Interactive effects of symbionts on microbial numbers

Bacterial density was significantly greater on average across treatments in the presence of yeast than in its absence (F-test of *modality* main effect: $F_{1,229}$ = 4.02, p = 0.0463). However, the effect was modest–approximately 0.18 orders of magnitude (± 0.09 standard error of the mean)–compared to among-strain variation–approximately 3 orders of magnitude (F-test of *bacterial treatment* main effect: $F_{19,229}$ = 11.07, p < 0.0001) (Fig 1A). Even though some bacterial strains seemed to benefit more from yeast presence than others, the interaction between the presence / absence of the yeast and the bacterial strain identity was not significant (F-test of *modality* * *bacterial treatment* interaction: $F_{19,229}$ = 0.86, p = 0.6280) (Fig 1B).

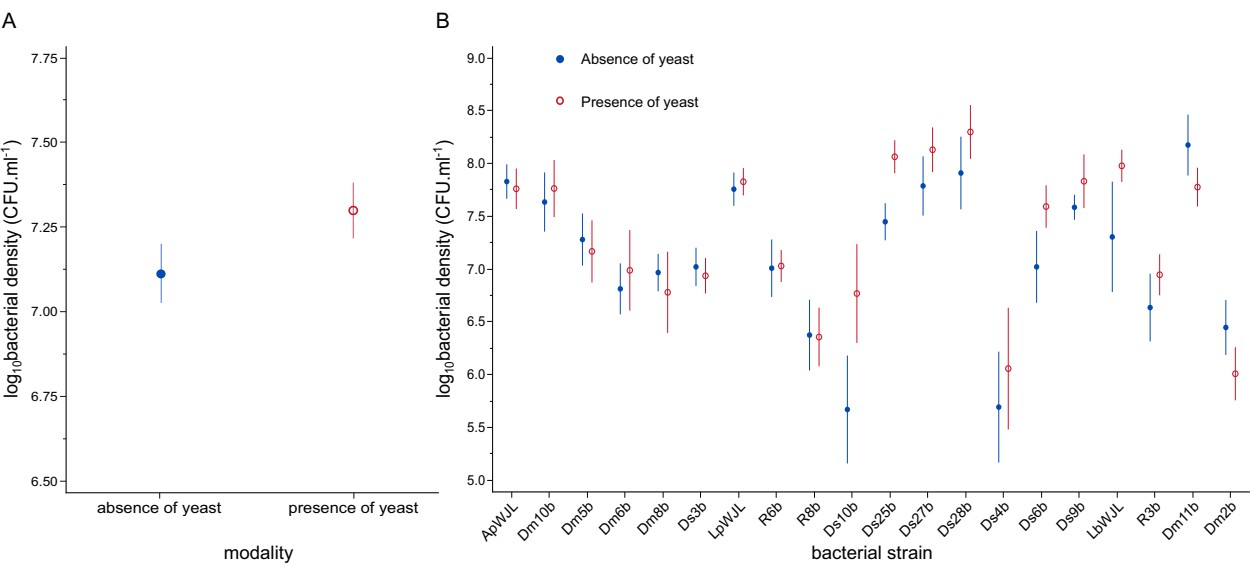

**Fig 1.** Effect of yeast on bacterial numbers in fruit across all bacterial strains (A) and by bacterial strain (B). Bacterial densities are expressed in $\log_{10}$ of colony forming units (CFU) per milliliter of fruit homogenate. At the time of inoculation, $\log_{10}$ bacterial density was 2.7 on the Y axis. Error bars indicate standard errors around the mean.

Yeast density was significantly influenced by bacterial strain ($F_{19,110} = 4.12$, $p < 0.0001$) (Fig 2A). Unfortunately, we did not measure yeast growth in the absence of bacteria. It was therefore not possible to assess whether bacteria had a generally beneficial or costly effect on the density of *H. uvarum* yeast. Overall, yeast and bacterial mean densities did not correlate significantly. Using Major Axis regression, the correlation coefficient between bacterial mean density and yeast mean density was 0.2198; the slope was 2.7458 (95% CI [-2.1477, 0.4665])

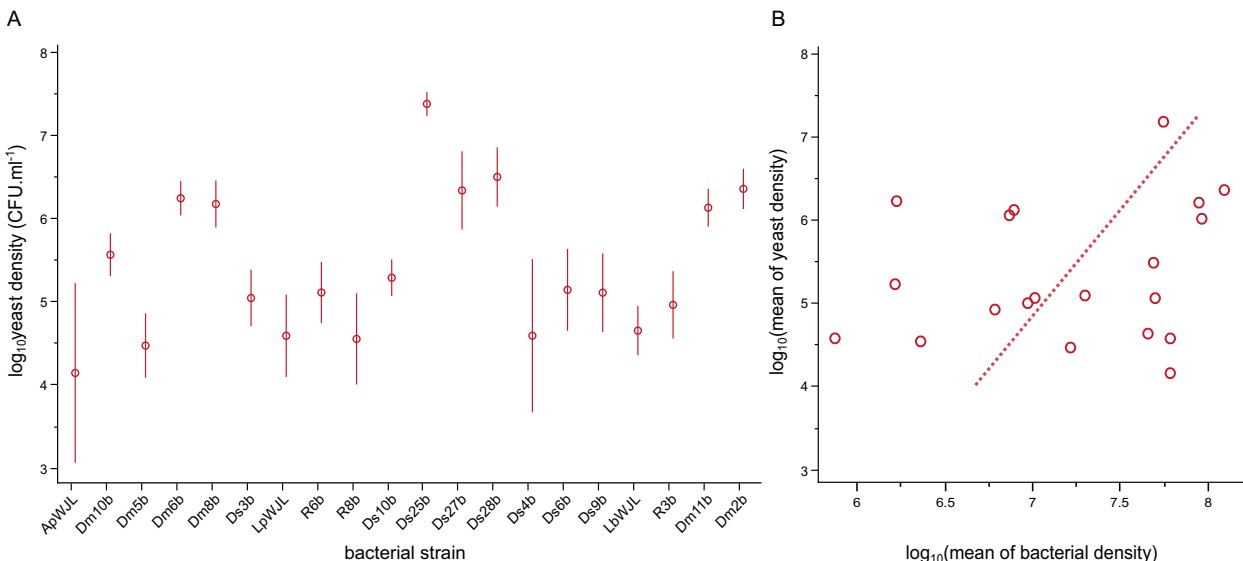

**Fig 2.** Effect of bacterial identity on yeast numbers in fruit (A) and Major Axis regression of $\log_{10}$(mean of yeast densities) on $\log_{10}$(mean of bacterial densities) (B). Yeast densities are expressed in $\log_{10}$ of colony forming units (CFU) per milliliter of fruit homogenate. In Fig 2A, at the time of inoculation, $\log_{10}$ yeast density was 2.7 on the Y axis. Error bars indicate standard errors around the mean. In Fig 2B, each point represents the mean yeast density and the bacterial density for a given bacterial strain.

(Fig 2B). Using OLS regression, the slope of yeast density as a function of bacterial density was 0.2847 (95% CI [-0.3411, 0.9104]) and the slope of bacterial density as a function of yeast density was 0.1697 (95% CI [-0.2033, 0.5426]), neither of which was significantly different from zero. Thus Major Axis regression and OLS regression consistently reported no correlation significantly different from zero.

### Relationships between cell numbers mean and variances

Overall, bacterial cell density across strains followed a spatial mixed-species TL. The slopes of the regression of log(variance bacterial density) on log(mean bacterial density) was 2.044 (95% CI [1.7601, 2.3284]) in the absence of yeast, and 2.062 (95% CI [1.7742, 2.3489]) in its presence. The intercepts and the slopes of the two linear regressions were not significantly different (F-test of *modality* main effect: $F_{1,36} = 1.56$, $p = 0.2199$; F-test of *modality*$^*$log$_{10}$*(mean)* interaction: $F_{1,36} = 0.01$, $p = 0.9290$) (Fig 3). The coefficient of the quadratic term '[log$_{10}$*(mean)*]$^2$' and the interaction of the quadratic term with yeast presence or absence were not significant (F-test of

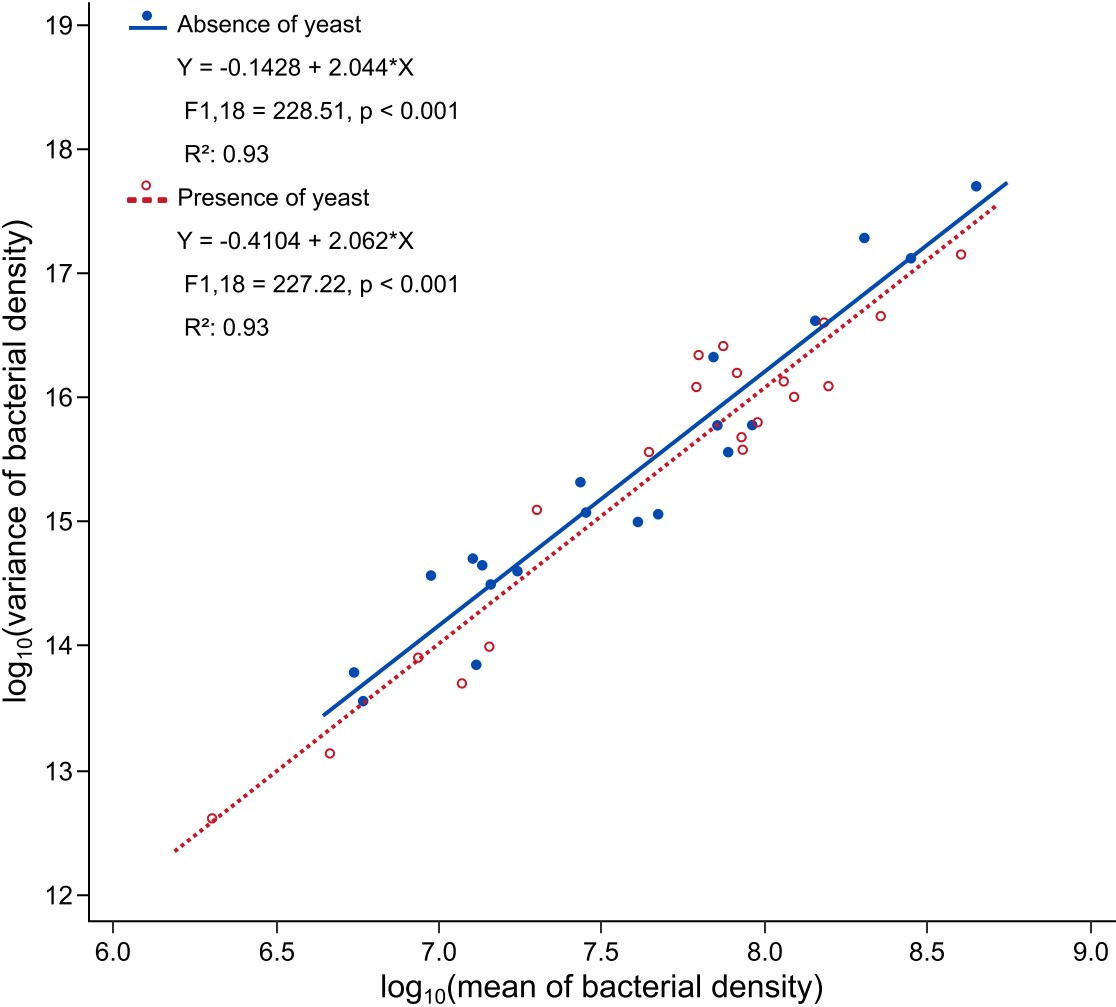

**Fig 3. Taylor's law holds in bacterial symbionts of *Drosophila*, and TL parameters are not significantly affected by yeast presence.** Linear regressions of log$_{10}$(variance) on log$_{10}$(mean) for the densities of each bacterial strain in the absence (solid blue line) and the presence (dashed red line) of yeast. Each point corresponds to one bacterial strain in the absence (blue filled circle •) or presence (red open circle o) of yeast.

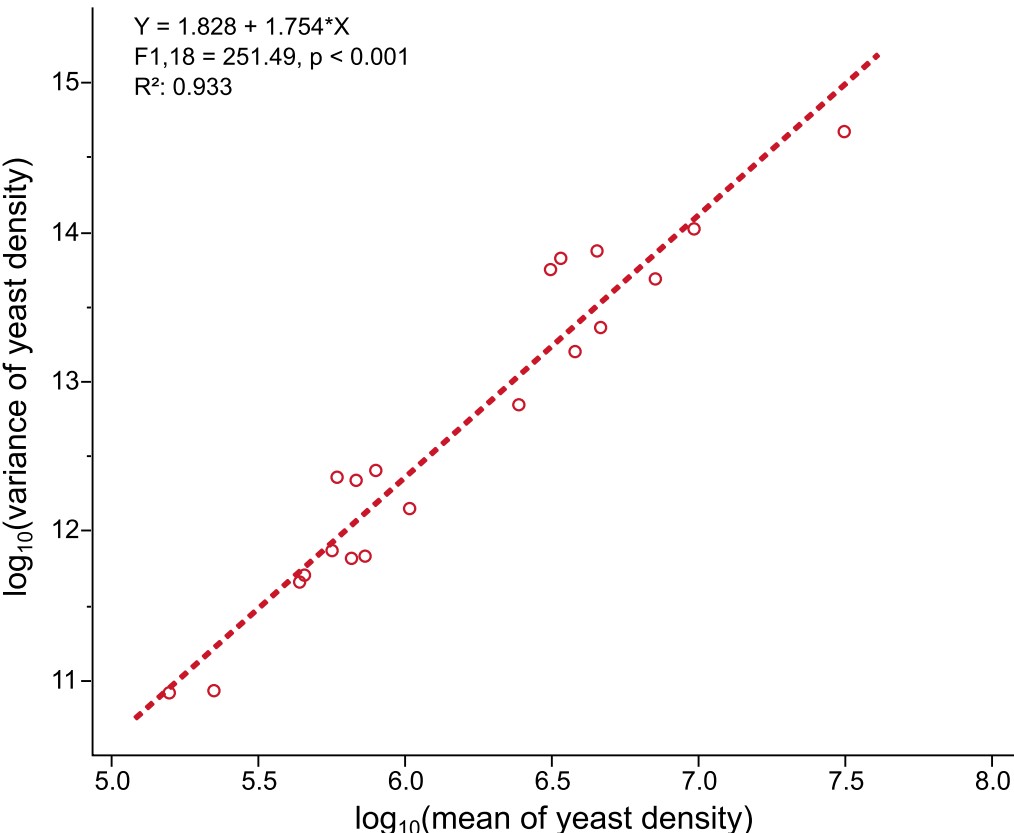

**Fig 4. Taylor's law holds in a *Hanseniaspora uvarum* yeast strain in presence of different bacterial symbionts.**
Linear regression of $\log_{10}$(variance of yeast density) on $\log_{10}$(mean of yeast density). Each point corresponds to a yeast density for a given bacterial strain.

$[\log_{10}(mean)]^2$ main effect: $F_{1,34} = 0.06$, $p = 0.8152$; F-test of *modality*\* $[\log_{10}(mean)]^2$ interaction: $F_{1,34} = 3.06$, $p = 0.0891$). The variance of bacterial density throughout the different microbial treatments therefore appeared to be consistent with a power function of mean bacterial density, as described by TL, and the parameters of TL were not affected by the presence of the yeast symbiont.

Taylor's law also appeared to hold in yeast densities (Fig 4). We found a linear positive relationship between $\log_{10}$(mean) and $\log_{10}$(variance) of yeast densities (F-test of $\log_{10}(mean)$ interaction: $F_{1,18} = 251.49$, $p < 0.001$). The slope of the regression was 1.828 (95% CI [0.3867, 3.2686]). The quadratic term '$[\log_{10}(mean)]^2$' was not significant (F-test of $[\log_{10}(mean)]^2$: $F_{1,17} = 2.58$, $p = 0.1267$).

## Discussion

We demonstrated interactions between *D. melanogaster*'s bacterial and yeast symbionts in ecologically realistic conditions. In our experiment, yeast presence slightly increased bacterial multiplication in fruit flesh infested with *Drosophila* larvae (Fig 1A). The twenty bacterial strains we tested also had variable effects on yeast multiplication (Fig 2). However, these interactive effects did not change the spatial Taylor's Law (TL): in the presence as in the absence of yeast, the variance of bacterial population density across the twenty bacterial strains was related to the mean of bacterial population density by a power law with a log-log slope indistinguishable from 2 (Fig 3).

Interactions between microbial symbionts emerge as important factors affecting microbial dynamics [31, 32]. However, the nature and prevalence of interactions between symbionts of *Drosophila* larvae in ecologically realistic conditions had never been investigated to our knowledge, in particular with wild microbial strains. Our study hence sheds light on novel aspects of *Drosophila* symbiosis in the field. We found that yeast apparently benefited bacteria, although modestly (Fig 1). Besides, we observed large variation in *Hanseniaspora uvarum* yeast cell densities as a function of the strain of bacterium with which it shared the symbiotic environment (Fig 2A). These interactive effects may be the result of cross-feeding, which is wide-spread in symbiotic systems. It is well-established that yeast associated to *Drosophila* produce ethanol that is converted to acetic acid by *Acetobacteraceae* bacteria, a phenomenon referred to sour-rot in farming [33, 34]. A similar cross-feeding interaction occurs between *Saccharomyces* yeast and *Lactobacillus* bacteria [35], which are among the most important bacterial members of the *D. melanogaster* microbiota [24, 26]. In our experiment *H. uvarum* yeast may have provided nutrients to most of the bacterial strains, even though indirect effects through host physiology cannot be ruled out [5, 36]. Independent of the mechanisms, interactive effects between symbionts may affect the dynamics of all partners at local and meta-population scales [37, 38].

Interactions between organisms and their density dependence may affect TL parameters, as genetics, ecology and other spatio-temporal factors do [10, 17, 20, 39–41]. Kilpatrick and Ives [42] predicted that the strength of competition between species would affect the slope in temporal versions of TL. A previous study tested a modification of this prediction (for spatial TL instead of temporal TL) with free-living bacteria that were grown alone or in competition in artificial environments of variable nutrient richness [9]. Though competition did occur between the tested bacteria, it did not change the slopes of the spatial TL from 2, with or without competition. Here, we pursued this investigation further with symbiotic microorganisms in ecologically realistic conditions that facilitated each other rather than competed. Our results showed the variance of bacterial population density in different replicates related to the mean of bacterial population density by a power law consistent with TL. As in the case of competition [9], we found no significant effect of facilitation on the form or parameters of a spatial TL (Fig 3). The discrepancy between our results for a spatial TL and Kilpatrick and Ives's predictions [42] about the effect of competition on a temporal TL highlights the importance of details in experimental tests of theoretical predictions and leaves open the challenge of finding experimental conditions suitable to test Kilpatrick and Ives's predictions.

Another theoretical study showed that biological replicate numbers affect the parameters of TL [43]. TL's slopes different from 2 may be undetectable if a stochastic multiplicative growth process in a Markovian environment (e.g. bacterial multiplication) is observed for a duration that exceeds the natural logarithm of the number of biological replicates. In our experiment, bacterial numbers were on average 1000 times that of the inoculum, implying that bacteria replicated at least ten times, since $2^{10} \approx 1000$. Cell divisions likely exceeded 10 cycles as most bacteria in a closed system quickly move from an exponential growth phase to a plateau phase, where bacterial density is constant because appearing cells and dying cells are in equal numbers. We had seven replicates, and a minimum of 10 replication cycles is larger than $\ln7 \approx 1.95$. The slopes indistinguishable from 2 that were estimated in our study, and those of Ramsayer et al. [9], may thus be statistical artifacts due to the long duration of the experiments relative to the number of replicates. Computing a power analysis with TL parameters from our study revealed the minimum number of replicates needed to observe a significant change in the intercept and the slope would have been approximately 100 and 20 000 respectively. Such replication is very challenging to reach in most experimental systems. It is possible that inter-specific interactions such as competition [9] and facilitation (the present study) affect TL's slope during relatively short time frames (e.g. a few hours for rapidly multiplying organisms

such as bacteria). Unveiling the ecological mechanisms that alter TL's parameters using microbial organisms remains an exciting area of research that will necessitate experimental designs adapted to the specific demographic features of microorganisms.

Our study explored interactions between microbial symbionts associated with *D. melanogaster* larvae in ecologically realistic conditions. Screening twenty natural isolates of bacteria in the presence and absence of a freshly collected strain of the yeast *H. uvarum*, we found yeast had a small facilitative effect (Fig 1A) that did not differ statistically among bacteria (Fig 1B). Bacterial identity had a large influence on yeast multiplication (Fig 2). These interactions between symbionts demonstrate this phenomenon in natural conditions in *D. melanogaster*, a key model system of symbiosis studies. Effects of yeast on bacteria did not significantly affect the parameters of the spatial version of TL. TL is a powerful means of investigating the forces that affect the demography of many species, however challenging it may be to use TL in microbial systems.

## Acknowledgments

We thank Guido Favia and two anonymous reviewers for comments on an earlier version of the manuscript. We thank Edouard Jurkevitch and Elodie Vercken for insightful comments. We thank Anne Xuéreb for her help at the beginning of the experiment and Laure Benoit for her help with the molecular identification of the bacterial strains.

## Author Contributions

**Conceptualization:** Robin Guilhot, Simon Fellous.

**Data curation:** Robin Guilhot, Joel E. Cohen.

**Formal analysis:** Robin Guilhot, Simon Fellous, Joel E. Cohen.

**Funding acquisition:** Simon Fellous.

**Investigation:** Robin Guilhot, Simon Fellous, Joel E. Cohen.

**Methodology:** Robin Guilhot, Simon Fellous, Joel E. Cohen.

**Project administration:** Robin Guilhot, Simon Fellous.

**Supervision:** Simon Fellous, Joel E. Cohen.

**Validation:** Simon Fellous, Joel E. Cohen.

**Writing – original draft:** Robin Guilhot, Simon Fellous, Joel E. Cohen.

**Writing – review & editing:** Robin Guilhot, Simon Fellous, Joel E. Cohen.

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
