## [Decision Letter · Decision Letter 0]

9 Jul 2020

PONE-D-20-19332

Yeast facilitates the multiplication of Drosophila bacterial symbionts but has no effect on the form or parameters of Taylor’s law

PLOS ONE

Dear Dr. Guilhot,

Thank you for submitting your manuscript to PLOS ONE. After careful consideration, we feel that it has merit but does not fully meet PLOS ONE’s publication criteria as it currently stands. Therefore, we invite you to submit a revised version of the manuscript that addresses the points raised during the review process.

In particular i invite you to respond to the concerns raised by referee 1

We look forward to receiving your revised manuscript.

Kind regards,

Guido Favia, Ph.D.

Academic Editor

PLOS ONE

Journal Requirements:

2. In your Methods section, please provide additional location information of the collection sites, including geographic coordinates for the data set if available.

3. In your Methods section, please provide additional information regarding the permits you obtained for the work. Please ensure you have included the full name of the authority that approved the collection sites access and, if no permits were required, a brief statement explaining why.

Reviewers' comments:

Reviewer's Responses to Questions

**Comments to the Author**

1. Is the manuscript technically sound, and do the data support the conclusions?

Reviewer #1: Partly

Reviewer #2: Yes

2. Has the statistical analysis been performed appropriately and rigorously? 

Reviewer #1: I Don't Know

Reviewer #2: Yes

3. Have the authors made all data underlying the findings in their manuscript fully available?

Reviewer #1: Yes

Reviewer #2: Yes

4. Is the manuscript presented in an intelligible fashion and written in standard English?

Reviewer #1: Yes

Reviewer #2: Yes

5. Review Comments to the Author

Reviewer #1: The manuscript by Guilhot and colleagues is very interesting and proposed a new aspect to look for in the interactions between microorganisms. Although the topic is very interesting, the study presents a few major weaknesses that should be addressed before the manuscript is suitable for publication.

General considerations:

Although the abstract, introduction and material and methods are well written, results should be more detailed while the discussion should be shortened.

Specific comments

* In the experimental design, at the end of the experiment, the authors sampled bacteria from the fruit pulp for cell counts. It is not clear how the authors ensured that the sampled microorganisms are the bacteria that they inoculated at the start of the experiment and not a consortium formed by their bacteria of interest and the endophytes already present in the fruit pulp. If the authors have taken measures to ensure the isolation of only the bacteria that they inoculated it would be important to add this step in the description of the experimental design. If the authors did not implement a protocol to isolate their bacteria nor to ensure that the microorganisms that they isolated are indeed the bacteria that they inoculated, there is a possibility that the bacterial densities that they measure are not those of the bacterial species they inoculated but of that of the cultivable bacterial community that was present in the fruit pulp at the end of the experiment. If this is the case both the results and the discussion should be modified to take into account this possibility.

* The a and b of Taylor’s power law [log(variance of population density)= a + b log(mean of population density)] are specific to the species but in this study, these two parameters are inferred globally for different species. The authors should recalculate the two parameters singularly for each species and include them in a result table and base their result description and discussion section on that table.

* In the figures 1A and 2B the data points represent independent results and should not be linked by a line.

* line 223-225: “In our experiment … behavior”, The experimental design is not able to assess the effect on the yeast due to the direct or indirect (bacteria interacting with the insect ) interactions with Drosophila especially since both bacteria and yeast are inoculated to the fruit pulp and later isolated from it. In this experiment, Drosophila represents only another variable present in all the experiments. In order to observe a potential effect due to the interaction with Drosophila, the authors should have planned for another series of experiment replicating the entire setup but without adding the fly.

* line 227-237: “Host-mediated … participate in”, since the experimental design does not allow to test the effects of the microorganisms on the fly nor does it allow to assess the effects of the insect on the microorganisms, this part should be removed since it is highly speculative in regards to the experiment.

* line 258-259: “In our experiment … since 2^10 = 1000.”, The inference that the authors had 10 generations based on the final number of bacteria will hold only if the bacteria are in an exponential phase of growth which is highly improbable after 7 days in a closed system (the petri dish). Some of the bacteria used in the experiment are known to have a short generation time and will probably reach their highest density in a short amount of time, after which their apparent growth will halt (plateau phase). In this phase, the number of new bacteria with bacteria multiplying but generations appearing but the total number of the population is constant because of the dying bacteria. Because of the dynamics of bacterial populations in a closed system, unless the bacteria are in an exponential growth phase it is very hard to determine the number of generations that have passed based on the final population size.

Reviewer #2: The article gives an insight of how the interactions between microbial symbionts may influence theirs and host's demography. The results show new prospectives on the relationship between Drosophila melanogaster and yest and bacteria.

6. PLOS authors have the option to publish the peer review history of their article (what does this mean?). If published, this will include your full peer review and any attached files.

Reviewer #1: No

Reviewer #2: No

---

## [Author Response · Author response to Decision Letter 0]

26 Oct 2020

Dear editor and reviewers,

We created a document that compiles your comments and our responses.

Please find attached this document.

---

## [Decision Letter · Decision Letter 1]

9 Nov 2020

Yeast facilitates the multiplication of Drosophila bacterial symbionts but has no effect on the form or parameters of Taylor’s law

PONE-D-20-19332R1

Dear Dr.  Guilhot,

We’re pleased to inform you that your manuscript has been judged scientifically suitable for publication and will be formally accepted for publication once it meets all outstanding technical requirements.

Kind regards,

Guido Favia, Ph.D.

Academic Editor

PLOS ONE

Additional Editor Comments (optional):

Reviewers' comments:

Reviewer's Responses to Questions

**Comments to the Author**

1. If the authors have adequately addressed your comments raised in a previous round of review and you feel that this manuscript is now acceptable for publication, you may indicate that here to bypass the “Comments to the Author” section, enter your conflict of interest statement in the “Confidential to Editor” section, and submit your "Accept" recommendation.

Reviewer #1: All comments have been addressed

Reviewer #2: All comments have been addressed

2. Is the manuscript technically sound, and do the data support the conclusions?

Reviewer #1: Yes

Reviewer #2: Yes

3. Has the statistical analysis been performed appropriately and rigorously? 

Reviewer #1: (No Response)

Reviewer #2: Yes

4. Have the authors made all data underlying the findings in their manuscript fully available?

Reviewer #1: Yes

Reviewer #2: Yes

5. Is the manuscript presented in an intelligible fashion and written in standard English?

Reviewer #1: Yes

Reviewer #2: Yes

6. Review Comments to the Author

Reviewer #1: The paper by Guilhot and colleagues has improved after addressing the different comment and should be considered for publication

Reviewer #2: In this article is reported the interactions between D. melanogaster’s bacterial and yeast symbionts in real conditions. The experiments showed how the yeast presence slightly increased bacterial

multiplication in fruit flesh infested by Drosophila without changing the spatial Taylor's law.

7. PLOS authors have the option to publish the peer review history of their article (what does this mean?). If published, this will include your full peer review and any attached files.

Reviewer #1: No

Reviewer #2: No

---

## [Editor Report · Acceptance letter]

13 Nov 2020

PONE-D-20-19332R1 

Yeast facilitates the multiplication of *Drosophila* bacterial symbionts but has no effect on the form or parameters of Taylor’s law 

Dear Dr. Guilhot:

I'm pleased to inform you that your manuscript has been deemed suitable for publication in PLOS ONE. Congratulations! Your manuscript is now with our production department. 

Kind regards, 

on behalf of

Prof. Guido Favia 

Academic Editor

PLOS ONE